# α-Fe_2_O_3_/TiO_2_/Ti_3_C_2_T_x_ Nanocomposites for Enhanced Acetone Gas Sensors

**DOI:** 10.3390/s24082604

**Published:** 2024-04-18

**Authors:** Zhihua Zhao, Zhenli Lv, Zhuo Chen, Baocang Zhou, Zhigang Shao

**Affiliations:** 1College of Mechanical and Electrical Engineering, Henan University of Technology, Zhengzhou 450052, China; lvzhenli@stu.haut.edu.cn (Z.L.); zhuo.chen@haut.edu.cn (Z.C.); keaiduo_2023@haut.edu.cn (B.Z.); 2Laboratory of Coordination Chemistry, CNRS UPR 8241, University of Toulouse, 205 Route de Narbonne, 31077 Toulouse, France

**Keywords:** gas sensors, Mxene, acetone response, α-Fe_2_O_3_, TiO_2_

## Abstract

Metal oxide semi-conductors are widely applied in various fields due to their low cost, easy processing, and good compatibility with microelectronic technology. In this study, ternary α-Fe_2_O_3_/TiO_2_/Ti_3_C_2_T_x_ nanocomposites were prepared via simple hydrothermal and annealing treatments. The composition, morphology, and crystal structure of the samples were studied using XPS, SEM, EDS, XRD, and multiple other testing methods. The gas-sensing measurement results suggest that the response value (34.66) of the F/M-3 sensor is 3.5 times higher than the pure α-Fe_2_O_3_ sensor (9.78) around 100 ppm acetone at 220°C, with a rapid response and recovery time (10/7 s). Furthermore, the sensors have an ultra-low detection limit (0.1 ppm acetone), excellent selectivity, and long-term stability. The improved sensitivity of the composites is mainly attributed to their excellent metal conductivity, the unique two-dimensional layered structure of Ti_3_C_2_T_x_, and the heterojunction formed between the nanocomposite materials. This research paves a new route for the preparation of MXene derivatives and metal oxide nanocomposites.

## 1. Introduction

Acetone is a common chemical compound in the medical industry. However, if its concentration reaches a certain value, it can cause several negative effects on humans, such as respiratory stimulation, vomiting, and spasms [1,2]. Acetone can also be used as a biomarker for diagnosing diabetes [3]. The principle is that the acetone concentration in the exhaled breath of healthy individuals is lower than that of those with diabetes [4]. People with diabetes exhale about 1.8–10 ppm of acetone gas, whereas people without diabetes exhale less than 0.8 ppm [5]. Therefore, the selective and fast detection of acetone below the critical ppm level is crucial for its industrial safety and the early diagnosis of diabetes. Gas-sensing technologies are an effective way to implement real-time gas detection; thus, this technology is often used to monitor air quality, food freshness, and human health [6]. In order to meet practical application requirements, gas sensors must have an excellent sensing performance, such as selectivity, sensitivity, and rapid response, which are closely related to sensing materials. Among them, semi-conductors are the most common due to their low cost, high processability, and good compatibility with microelectronic technology [7,8]. Therefore, many metal oxide semi-conductors have been used as gas-sensing materials for acetone vapor detection [9,10,11,12].

According to previous works, α-Fe_2_O_3_ has an ideal gas sensitivity level for methyl-containing gas, especially acetone [13,14]. However, pure α-Fe_2_O_3_ gas sensors generally have low responses and relatively high detection limits for acetone, which limits its application in human health diagnoses. To overcome these limitations, heterojunction, composite materials, and elemental doping, etc., have been attempted. For example, Lu et al. successfully prepared a W_18_O_49_/α-Fe_2_O_3_ hollow heterojunction structure where the acetone detection limit of the material reached 86 ppb at 260 °C, which is much lower than the detection limit of pure α-Fe_2_O_3_ [15]. Wang et al. reported that the microstructure of α-Fe_2_O_3_ was changed by Ce doping and generated surface defects created more active sites for gas adsorption. Therefore, the 100 ppm acetone response can reach 26.3 at 220 °C [16]. Liu et al. mentioned that the response of α-Fe_2_O_3_ with a porous structure of up to 100 ppm acetone at 210 °C can reach 14.5, indicating that the porous structure of α-Fe_2_O_3_ has a low detection limit and a short response time [17]. Guo et al. reported that the response of rGO/α-Fe_2_O_3_ nanofibers to 100 ppm acetone can reach 100, which is about 4.5 times that of the response of a α-Fe_2_O_3_ nanobelt at 375 °C [18]. Although α-Fe_2_O_3_ gas sensitivity has been improved via these methods, further study is required to obtain an acetone sensor with both low detection limits and a rapid response.

MXene is a novel two-dimensional material that has gained significant attention for its widespread applications in the fields of photo/electro catalysts, supercapacitors, electronic ink, electromagnetic wave shielding, and gas sensors [19,20,21]. For example, monolayer Ti_2_CO_2_ MXene has shown an outstanding NH_3_ sensing ability due to its unique transport properties. In addition, its conduction characteristics can dramatically change before and after NH_3_ adsorption [22]. Ti_3_C_2_T_x_ MXene films, developed on polyimide substrates via a simple drop-casting method, detected various VOCs, such as CH_3_OH, C_2_H_5_OH, and NH_3_ at room temperature [23,24]. In another work by Lu et al., a simple alkaline treatment improved the gas response of Ti_3_C_2_T_x_ MXene, possibly due to the intercalation of an alkali metal (Na^+^) ion [25]. Moreover, Ti_3_C_2_T_x_ MXene has a large surface area and rich surface groups, making it a good substrate when compounded with other materials. For example, different composites of Ti_3_C_2_T_x_ MXene, such as PANI/Ti_3_C_2_T_x_ [26], Pd@MXene [27], W_18_O_49_/Ti_3_C_2_T_x_ [28], In_2_O_3_ nanofibers/Ti_3_C_2_T_x_ MXene [29], α-Fe_2_O_3_/Ti_3_C_2_T_x_ [30], and CuO/Ti_3_C_2_T_x_ [31], have been prepared for advanced gas sensors. However, no reports have been published on the ternary α-Fe_2_O_3_/TiO_2_/Ti_3_C_2_T_x_ nanocomposites for gas sensor applications.

In this ongoing study, ternary α-Fe_2_O_3_/TiO_2_/Ti_3_C_2_T_x_ nanocomposites were prepared using simple hydrothermal and annealing treatments. In addition, the materials were tested for their gas sensitivity, selectivity, moisture resistance, and stability. The results show that the sensors’ optimal response to 100 ppm acetone was 34.66. The sensors also showed great selectivity and a lower detection limit (0.1 ppm acetone), which is important for theoretical applications.

## 2. Materials and Methods

Ferr (III) chloride hexahydrate, ammonia, and hydrofluoric acid (HF) were purchased from Aladdin. Acetone, ethanol, methanol, formaldehyde, and toluene were purchased from Sinopharm Chemical Reagent Co., Ltd., Shanghai, China. The above reagents were of an analytical grade and did not require further purification. Ti_3_AlC_2_ (98%) powder was obtained from Jilin 11 Technology Co., Ltd., Jilin, China.

### 2.1. Synthesis of Ti_3_C_2_T_x_ MXene

The Ti_3_C_2_T_x_ sample was prepared using the methods mentioned in our previous work [32]. First, 1g of Ti_3_C_2_T_x_ was added to 20 mL of 45 wt % HF. Then, the solution was stirred at room temperature for 24 h. The solution was washed with deionized water until the pH value was about 6~7. Finally, the collected precipitate was dried at 80°C for 12 h in a vacuum to obtain Ti_3_C_2_T_x_ powder.

### 2.2. Synthesis of α-Fe_2_O_3_

The α-Fe_2_O_3_ was synthesized via simple hydrothermal and annealing treatment. Under normal circumstances, 677 mg of FeCl_3_·6H_2_O (AR, Aladdin Reagent, Shanghai, China) was added to 20 mL of deionized water. After it was dissolved, 2 mL of NH_3_·H_2_O (AR, Aladdin Reagent, Shanghai, China) was added dropwise into the aforementioned solution, followed by stirring with a magnetic force for 1 h. As a result, the emulsion was transferred into a PTFE autoclave, and then was heated at 150 °C for 12 h. The precipitate was collected by washing repeatedly with deionized water and absolute ethanol, and then was dried in a vacuum at 80 °C for 12 h. Finally, the precipitate powder was annealed at a rate of 10 °C/min for 30 min 350 °C to obtain α-Fe_2_O_3_ powder.

### 2.3. Synthesis of α-Fe_2_O_3_/TiO_2_@Ti_3_C_2_T_x_ Nanocomposites

The synthesis process used for α-Fe_2_O_3_/TiO_2_/Ti_3_C_2_T_x_ nanocomposites was similar to that of pure α-Fe_2_O_3_. The procedure was as follows. First, 20 mL of deionized water was added to 677 mg of FeCl_3_·6H_2_O to obtain an orange solution. Then, 6.19 mg of Ti_3_C_2_T_x_ powder was slowly added to the aforementioned solution and the solution was stirred until it was uniformly dispersed. Then, 2 mL of NH_3_·H_2_O was added dropwise to the solution and stirred magnetically at room temperature for 1 h. The solution was then transferred to a PTFE pressure cooker and heated at 150 °C for 12 h. The sediment was collected after repeated washing with deionized water and anhydrous ethanol, and then the collected precipitate was kept in a vacuum at 80 °C for 12 h. Finally, the powder was calcinated at 350 °C with a temperature increase of 10 °C/min for 30 min to obtain the ternary α-Fe_2_O_3_/TiO_2_/Ti_3_C_2_T_x_ nanocomposites. The weight proportions of 2.02, 4.082, 6.19, and 8.33 mg of Ti_3_C_2_T_x_ within the nanocomposites were theoretically calculated to be 1%, 2%, 3%, and 4%, and thus, the nanocomposites with four different proportions were prepared using the same procedure. For convenience, the final products were marked as F/M-1, F/M-2, F/M-3, and F/M-4, respectively.

### 2.4. Structures and Morphology Characterizations

The X-ray powder diffraction (XRD) patterns of the obtained samples were obtained via an X-ray diffractometer (XRD, X’ PertPro MPD, PANalytical BV, Holland) with Cu Kα radiation (λ = 1.5442 Å). The crystal property data were confirmed using the selected 2θ range of 5–90°. Scanning electron microscopy (SEM, Gemini SEM 300, ZEISS, Germany) characterization was conducted to observe the morphology of the materials. The elemental mappings were acquired using energy disperse X-ray spectroscopy (EDS, Oxford Instrument, England). The surface composition and electronic state of the samples were analyzed using X-ray photoelectric spectroscopy (XPS, PHI-5300, Perkin Elmer, USA) with a monochromatic Al Kα source (1486.6 eV).

### 2.5. Evaluation of Gas-Sensing Performance

The method used for preparing the sensors and evaluating their performance was similar to the method mentioned in our previous work [32]. First, an appropriate amount of the nanocomposites was added to a suitable amount of anhydrous ethanol, and the mixture was sonicated for 2–5 min until the nanocomposites were completely dissolved. It was confirmed that the electrode on the ceramic tube was completely coated in the material and the thickness of the paint was uniform. The coated ceramic tube was vacuum-dried at 80 °C for 6 h. Then, the dried ceramic tube was welded to the base. Subsequently, to control the temperature of the sensitive layer, a heated Ni-Cr alloy coil was passed through the ceramic tube and welded to the base. The gas sensitivities were tested using the WS-30B gas sensitive element test system (Zhengzhou Wisen Electronic Technology Co., Ltd., Zhengzhou, China); the exact procedure is shown in Figure 1. The operating temperature of the gas sensor was adjusted by changing the heating voltage. All the tested gases were liquid at room temperature, the liquids were transferred onto a heating plate installed in the testing chamber, and evaporation was used to obtain the corresponding testing gases. The relative humidity of the testing chamber could be adjusted roughly by adding deionized water to the evaporator to obtain water vapor. Here, all the target gases were reducing gases, and the resistance of the sensor decreased when exposed to a tested gas. Thus, the gas-sensing response was defined as R_a_/R_g_, where R_a_ and R_g_ represent the resistance of the gas sensor in the air and target gas, respectively. The response and recovery time was defined as the time taken to reach a 90% change in the resistance during the gas adsorption and desorption processes, respectively.

## 3. Results

### 3.1. Structures and Morphology of Samples

The phase and crystal structures of the samples were investigated via a series of X-ray diffraction (XRD) analyses. Figure 2a shows the XRD spectra of the Ti_3_AlC_2_ MAX precursor and the obtained Ti_3_C_2_T_x_ MXene. Slight shifts in the (002) and (004) peaks of the Ti_3_C_2_T_x_ toward the lower angles are evident, while the most intense diffraction peak at 39° disappeared completely compared to the Ti_3_AlC_2_ MAX precursor, which indicates the successful conversion of Ti_3_AlC_2_ into Ti_3_C_2_T_x_ [33]. Figure 2b shows the XRD patterns of the α-Fe_2_O_3_ and F/M composites samples. The diffraction peak of the obtained pure α-Fe_2_O_3_ matches well with the standard card (JCPDS No. 33-0664), which reveals that the α-Fe_2_O_3_ was synthesized successfully. However, there were no significant Ti_3_C_2_T_x_ diffraction peaks for the F/M-1, F/M-2, F/M-3, and F/M-4 samples, which may have been due to the relatively low content of Ti_3_C_2_T_x_ and the α-Fe_2_O_3_ coverage produced in the annealing process [28].

The morphologies and microstructures of the Ti_3_C_2_T_x_ and F/M-3 nanocomposites were studied using SEM. As shown in Figure 3a, the obtained Ti_3_C_2_T_x_ shows an accordion-like structure. The SEM images with different resolutions (Figure 3b,c) of F/M-3 indicate that the layered structure of Ti_3_C_2_T_x_ was well-preserved and can be used as a substrate to support α-Fe_2_O_3_ nanoparticles. The EDS element mapping results are shown in Figure 3d, confirming the uniform distribution of the α-Fe_2_O3 nanoparticles on the layered MXene. This structure endowed the sample with a large specific surface area. Ti_3_C_2_T_x_ can also be used as a carrier for charge transfer, so the gas-sensing performance was improved. Figure 4 shows the XPS analysis results of the F/M-3sample. As shown in Figure 4a, peaks conforming to Ti, C, O, F, and Fe were broadly observed, which are in agreement with the EDS results. It can be seen that the intensities of the Ti and C energy spectra are much lower than that of the Fe energy spectrum, indicating the low content of Ti_3_C_2_T_x_. Figure 4b shows the energy spectrum of Fe. It can be seen that the combined binding energy at 712.2 and 724.1 eV, respectively, represent the peak values of Fe 2p3/2 and Fe 2p1/2, indicating that Fe may exist as a form of α-Fe_2_O_3_ in the F/M nanocomposites [34]. In Figure 4c, the C 1s XPS clearly shows that the peak of the Ti-C bond of Ti_3_C_2_T_x_ at 281.7 eV vanished entirely in F/M-3. Additionally, the Ti 2p XPS (Figure 4d) of F/M-3 shows that the peak values of Ti^2+^ and Ti^3+^ (453~457 eV) decreased, while the peaks of Ti^4+^ (~458 eV) obviously increased compared to Ti_3_C_2_T_x_, indicating that the sectional Ti_3_C_2_T_x_ was oxidized into TiO_2_ [35]. The above characteristic analysis confirms that the synthetic samples consisted of α-Fe_2_O_3_, TiO_2_, and Ti_3_C_2_T_x_.

### 3.2. Gas-Sensing Performance

In Figure 5a, the gas-sensing responses of the different gas sensors to 100 ppm acetone under operating temperatures between 180 °C and 260 °C can be seen. Regardless of the working temperature, all F/M nanocomposite-based sensors showed substantially greater response values than the pure α-Fe_2_O_3_ sensors, and the sensor with the F/M-3 ratio had the highest response among all sensors. Additionally, it was found that the maximum response value of all gas sensors occurred at 220 °C. Figure 5b exhibited the dynamic response and recovery curves of sensors exposed to acetone ranging from 5 to 100 ppm at the optimal working temperature of 220 °C. Clearly, the response significantly increased with the increasing acetone concentration. Additionally, the strong ability of the response curve to return to its original position after acetone removal indicates a high degree of reversibility. Meanwhile, it can be seen in Table 1 that the response recovery times of F/M-3 to acetone at different concentrations were basically the same, which proves the response stability of the sensor. Further research was carried out on the sensors’ acetone gas detection capabilities. Figure 5c describes the reproducibly dynamic response and recovery curves of the pure α-Fe_2_O_3_ and F/M-3 sensors to ultra-low acetone concentrations of 0.1~3 ppm at 220 °C. The F/M-3 sensor showed a highly superior response compared to the pure α-Fe_2_O_3_ sensor when exposed to 0.5 ppm of acetone or more. Although the response of the F/M-3 sensor showed a drift from the baseline, the difference in the response value was negligible. This was likely due to the water vapor from the acetone gas. Because the low concentrations were outside of the minimum range of the micro-syringe that was used to transfer the liquid acetone to the evaporator, the original liquid acetone had to be properly diluted. When the water vapor came into contact with the alkalized Ti_3_C_2_T_x_ within the F/M-3 composite, the resistance of the alkalized Ti_3_C_2_T_x_ decreased [25], which decreased the resistance of the composite, leading to an increase in the R_a_/R_g_. This ultra-low detection limit makes the F/M-3 sensor suitable for applications such as diabetes diagnostics. As illustrated in Figure 5d, 100 ppm acetone, ethanol, methanol, formaldehyde, and toluene were used to study the selectivity of F/M-3. The response of acetone at different concentrations was significantly greater than that of the other four gases, indicating that acetone has good selectivity. The selectivity of acetone could be attributed to the different bond energies within the target gases. The bond disaggregation energy of acetone (366 kJ mol^−l^) is lower than that of gases such as formaldehyde (368 kJ mol^−1^), ethanol (462 kJ mol^−1^), and so on, so it reacts more easily with the adsorbed oxygen species. Moreover, the rich surface functional groups of MXene material and iron oxide may have a synergistic effect, which makes it easier for them to form strong hydrogen bonds with acetone molecules [30]. It should be noted that all the gas-sensing measurements above were conducted in an ambient environment with 10% relative humidity. Subsequently, the response change of the F/M-3 sensor under different relative humidities (RH = 10~50%) was further investigated. As shown in Figure 5e, the response of the F/M-3 sensor to 100 ppm acetone also steadily decreased as the RH increased, and the lowest value occurred when the RH was 50%. We believe that the F/M-3 test response decreased with increasing humidity at the optimal temperature, which can be explained by the fact that H_2_O molecules can compete with O_2_ molecules to capture electrons from the F/M-3 composite, which is not conducive to the reaction between the ionic oxygen species (O_2_^−^, O^−^, and O_2_^−^) adsorbed on the surface of the composite and the target gas. Therefore, the variation in resistance of the F/M-3 sensor decreased with increasing RH. Figure 5f shows the response and recovery times of the F/M-3 sensor to 100 ppm acetone at 220 °C, which were 10 s and 7 s, respectively. Additionally, the repeatability and long-term stability of the F/M-3 sensor to acetone was also investigated and are shown in Figure 6. The sensitivity value changed very little over five consecutive cycles, which demonstrates great repeatability (Figure 6a), and the response value of the sensor made out of ternary nanocomposites was quite stable. As shown in Figure 6b, over 47 days, the response to 100 ppm acetone is negligible. As illustrated in Table 2, compared to previous work on acetone sensors, the F/M composite sensor has a promising application for detecting acetone.

## 4. Discussion

The gas-sensitive mechanism of semiconductors was studied though the change of the resistance before and after adsorbing the target gas [41]. It is universally known that α-Fe_2_O_3_ is a typical n-type semiconductor, and thus, the exposed surface of the material can adsorb oxygen molecules in the air. Simultaneously, the electrons transmitted to the band can be extracted from oxygen atoms with strong electronegativity to form O_2_^−^, O^−^, and O_2_^−^ on the basis of the temperature, causing the formation of electron depletion layers and an increase in resistance. In addition, the operating temperature affects the type of adsorbed oxygen formation, and thus, a reaction of the adsorbed oxygen with the target gas. In general, O_2_^−^ is the main oxygen species when the temperature is below 150 °C. The O_2_^−^ species disappears rapidly when the temperature increases to between 150 and 400 °C, and O^−^ becomes the dominant oxygen species. When the operating temperature increases further than above 400 °C, O^2−^ is formed [42]. When the sensor is exposed to acetone gas, the acetone molecules will react with the oxygen to produce H_2_O and CO_2_, according to Equations (1)–(4). The released electrons returned back to conductive band, which reduces the depletion layers and increases the charge carrier density, leading to the decreased resistance of α-Fe_2_O_3_-based gas sensors.
O_2_(gas) → O_2_ (ads)(1)
O_2_(ads) + e^−^ → O_2_^−^(ads) (T < 150 °C)(2)
O_2_^−^(ads) + e^−^ → 2O^−^(ads) (150 °C < T < 400 °C)(3)
CO(CH_3_)_2_(ads)+ 8O^−^(ads)→ 3CO_2_(gas) + 3H_2_O (gas)+ 8e^−^(4)

The effect of Ti_3_C_2_T_x_ MXene on improving acetone gas-sensing performance is explained below. Based on the layered structure of Ti_3_C_2_T_x_ MXene, F/M nanocomposites may have larger specific surface areas, which can offer rich active sites for the adsorption of oxygen and acetone gas. Thus, the acetone gas sensitivity performance of the F/M nanocomposites was improved. But when the content of Ti_3_C_2_T_x_ in the nanocomposite was greater than that of the F/M-3 sample, the sensing performance decreased, which may have been due to the accumulation of materials caused by the excess Ti_3_C_2_T_x_, which, thus, reduced the specific surface area, resulting in a decrease in the active site.

The improved sensing performance was also due to the possible formation of ohmic contacts and n-n heterojunctions between Ti_3_C_2_T_x_ and its derivatives, TiO_2_ and α-Fe_2_O_3_ nanoparticles. It has been proven that the heterostructure can inhibit the recombination of carriers and increase the concentration of carriers, and thus, the gas-sensing performance of semiconductors has been greatly improved. The adsorption and desorption process of the gas is illustrated in Figure 7a. Compared to the α-Fe_2_O_3_ sensor, the F/M nanocomposite sensors exhibited a significantly superior response to acetone, which could probably be attributed to the construction of the α-Fe_2_O_3_/TiO_2_ n-n junction combined with metallic Ti_3_C_2_T_x_. As shown in Figure 7b, α-Fe_2_O_3_, TiO_2_, and Ti_3_C_2_T_x_ had different band structures before coming into contact with any species [34,43]. A carrier transfer process occurred after contact until the system reached equilibrium and a new Fermi level was obtained, resulting in the bands bending. Within the F/M nanocomposites, metallic Ti_3_C_2_T_x_ and α-Fe_2_O_3_ had a lower and higher work function, respectively, compared to TiO_2_. The results show that the electrons migrated from Ti_3_C_2_T_x_ and α-Fe_2_O_3_ to TiO_2_ and vice versa. This led to a loss layer at the n-n junction interface and the formation of TiO_2_@Ti_3_C_2_T_x_ formation of electron accumulation layers (EALs) on the interface, as shown in Figure 7c. In air, the O_2_ molecules absorbed on the surface caught electrons to form O_2_^−^, O^−^, and O^2−^ species. The results were a thinner EAL and thicker layers which were depleted, which decreased the charge carrier density and compressed the charge carrier mobility. Therefore, the resistance of the materials increased, as shown in Figure 7d. When exposed to acetone (Figure 7e), the electrons generated by the reaction between the acetone and oxygen species were released into the nanocomposites. In this case, the EALs and depleted layers were recovered at the original location, leading to the decrease in the resistance of the F/M sensors. Due to the existence of the ohmic contact and n-n heterojunction, the F/M sensors exhibited a greater transformation in resistance than the pure α-Fe_2_O_3_ sensor under the acetone atmosphere.

## 5. Conclusions

In summary, α-Fe_2_O_3_/TiO_2_/Ti_3_C_2_T_x_ ternary nanocomposites were successfully prepared using simple hydrothermal and annealing treatments. The gas sensitivity of the materials was evaluated carefully, and the results show that the gas sensitivity of the F/M composites was greater than that of pure α-Fe_2_O_3_. Additionally, gas sensors with an F/M-3 doping ratio exhibited the best performance, with a response value of up to 34.66 for 100 ppm of acetone at 220 °C, which is an increase of approximately 3.5-fold compared to pure α-Fe_2_O_3_. The sensor had a fast response and recovery (10 and 7 s, respectively, to 100 ppm of acetone), commendable selectivity, and a low detection limit for 0.1 ppm. The main reason for its improved sensitivity is the unique topography of F/M nanocomposites and the formation of heterojunctions between different components. In addition, the F/M-3 sensor has good moisture resistance and long-term stability, which has broad prospects in many practical applications, for instance, diabetes detection.

## Figures and Tables

**Figure 1 sensors-24-02604-f001:**
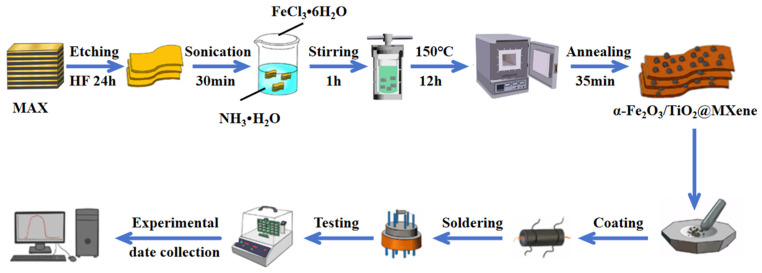
A flowchart of the experimental methodology framework.

**Figure 2 sensors-24-02604-f002:**
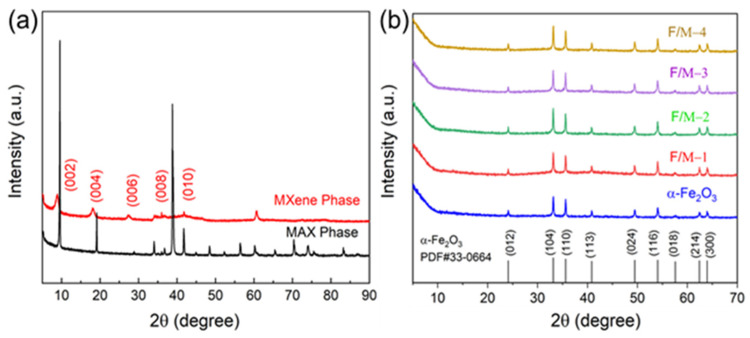
(**a**) XRD patterns of Ti_3_AlC_2_ and Ti_3_C_2_T_x_. (**b**) XRD patterns of the nanocomposites.

**Figure 3 sensors-24-02604-f003:**
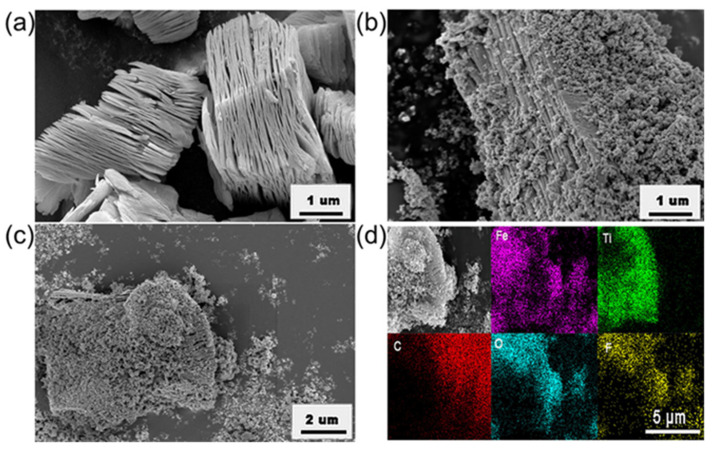
SEM images of (**a**) MXene, and (**b**,**c**) F/M-3 nanocomposites. (**d**) EDS mappings of F/M-3 nanocomposite.

**Figure 4 sensors-24-02604-f004:**
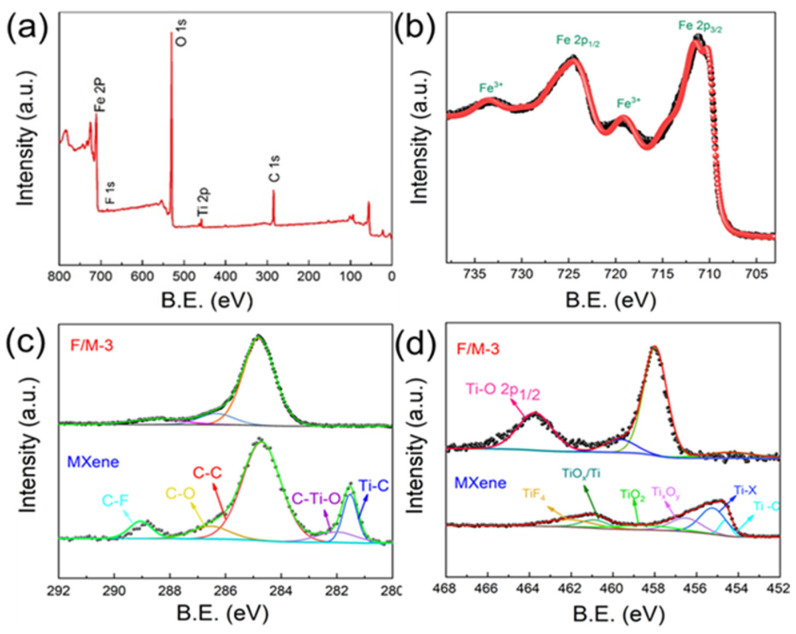
(**a**) Survey XPS spectra of F/M-3 nanocomposites. (**b**) Fe 2p spectra of F/M-3 nanocomposites. (**c**) XPS C 1s and (**d**) Ti 2p spectra of F/M-3 and pure MXene.

**Figure 5 sensors-24-02604-f005:**
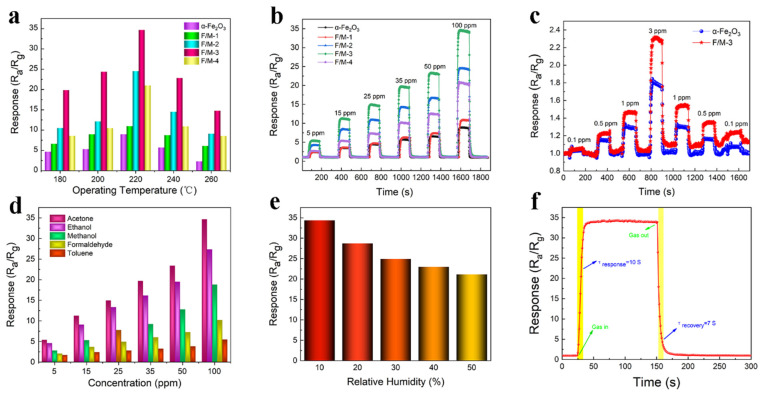
(**a**) The response of all samples to 100 ppm of acetone at different temperatures. (**b**) The transient response to 5–100 ppm of acetone. (**c**) The transient response of pure α-Fe_2_O_3_ and F/M-3 sensors to acetone at relatively low concentrations. (**d**) The selectivity of F/M-3 at different concentrations at 220 °C. (**e**) The response of the F/M-3 sensor to 100 ppm of acetone at different levels of humidity. (**f**) The response and recovery times of the F/M-3 sensor to 100 ppm of acetone at 220 °C.

**Figure 6 sensors-24-02604-f006:**
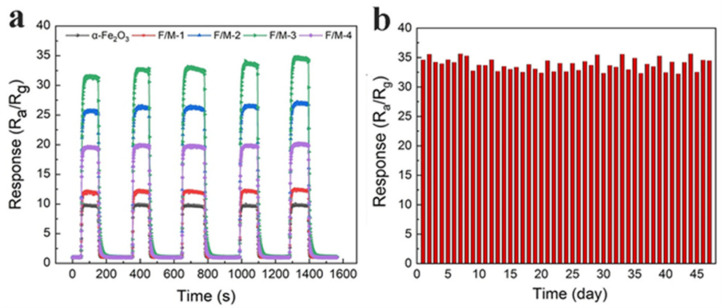
The reproducibility (**a**) and long-term stability (**b**) of the F/M-3 sensor to 100 ppm of acetone at 220 °C.

**Figure 7 sensors-24-02604-f007:**
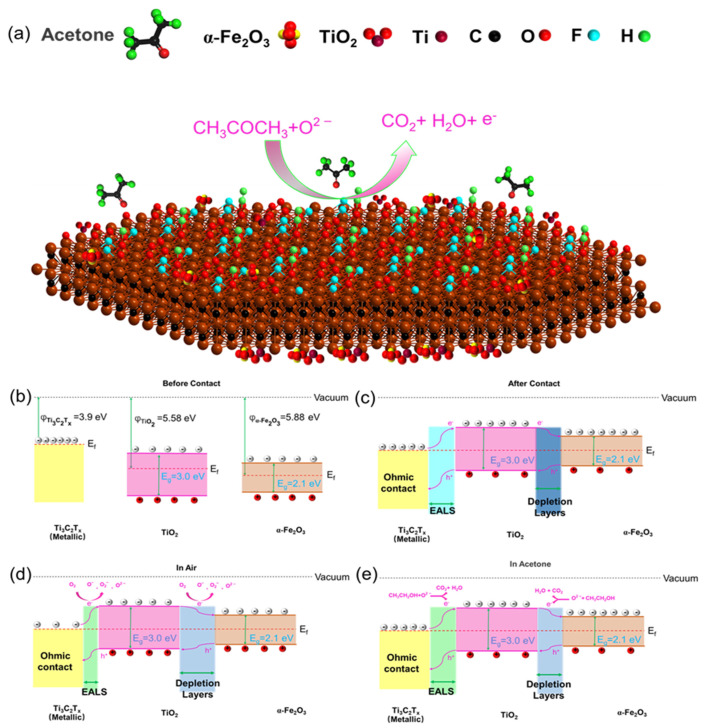
(**a**) Schematic of the reaction between acetone and F/M nanocomposites. (**b**–**e**) Schematic diagram of the band structure of the F/M nanocomposites.

**Table 1 sensors-24-02604-t001:** Response and recovery times of F/M-3 to different concentrations of acetone at 220 °C.

Concentration	5 ppm	15 ppm	25 ppm	35 ppm	50 ppm	100 ppm
Response time (s)	17	11	10	9	10	9
Recovery time (s)	9	9	10	10	12	9

**Table 2 sensors-24-02604-t002:** Comparison of gas-sensing performance between our results and those reported in the previous literature.

Sensing Material	Acetone (ppm)	T Sens (°C)	Response	Res/Rec (s/s)	Ref.
α-Fe_2_O_3_	100	340 °C	9.1	/	[36]
rGO/α-Fe_2_O_3_	100	225 °C	13.9	/	[37]
ZnSnO_3_/ZnO/Ti_3_C_2_T_X_	100	120	15.68	5/12	[38]
MXene/SnO_2_ heterojunctions	50	23.5	0.8%	/	[39]
Partially oxidized Ti_3_C_2_T_x_	2	350	180%	/	[35]
Metallic Ti_3_C_2_T_x_	100	RT	0.9~0.1%	/	[40]
α-Fe_2_O_3_/TiO_2_@Ti_3_C_2_T_x_ nanocomposites	100	220	34.66	10/7	This work

## Data Availability

Data are contained within the article.

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
