# Peer review of "α-Fe2O3/TiO2/Ti3C2Tx Nanocomposites for Enhanced Acetone Gas Sensors"

_sensors, 2024, doi:10.3390/s24082604_

Round 1

Reviewer 1 Report

Comments and Suggestions for Authors

In this work, authors reported the successful synthesis of α-Fe2O3/TiO2/Ti3C2Tx ternary nanocomposites, and characterized the morphology and microstructure by various characterizations. The acetone-sensing properties of as-synthesized samples were also measured and obtained the expected results. After carefully evaluate this manuscript, I would recommend for minor revision and authors should address the following points:

(1) Please write more experiment data in the abstract and conclusion to highlight the innovation of the research work.

(2) The thickness of the sensing film is a critical parameter. What is the thickness of the sensing film? Please provide the SEM cross section to confirm the thickness sensing film.

(3) The authors should provide more discussions on reasons for the observed selectivity. Reasons why this sensor shows high response to acetone gas have to be more discussed.

(4) Some related works about MXene-based gas sensors should be included as the references, such as Sens. Actuators B Chem. 409 (2024) 135541 and Sens. Actuators B Chem. 405 (2024) 135338.

Comments on the Quality of English Language

Minor editing of English language required.

Reviewer 2 Report

Comments and Suggestions for Authors

A manuscript is submitted for review, in which the preparation of a composite material based on Ti3C2Tx-maxene, titanium oxide and iron oxide is studied.

The work is interesting, however, it contains serious drawbacks. 

Main remarks:

- The abstract contains insufficient research on acetone-sensitive materials. In particular, the results of the responses of maxene-based acetone sensors that are sensitive to acetone concentrations of 50ppb and at room temperature are not shown. See, for example, 10.1021/acsanm.9b02223 ; [21]; ACS nano (2018) 12 No. 2: 986-993; S&A B (2021) 329:129275; ACS Sensors(2019) 4 â„– 6: 1603-1611.

- In the Discussion section, it is necessary to provide a table comparing the results obtained in the manuscript with the results in these and other articles.

- the authors have shown a significant effect of simultaneous exposure to moisture and acetone. However, they did not conduct studies on exposure to moisture and other gases, as well as long-term exposure to moisture and acetone.

Remarks:

L19-20. The phrase must be moved to the introduction.

L41-56. The authors pointed out the high operating temperature of acetone sensors based on iron oxide. However, there are known works in which Ti3C2Tx-maxene-based gas sensors are sensitive to acetone at room temperature. The authors did not mention this. These works should be studied and presented in the abstract. And in the "Discussion" section, it is necessary to provide a table comparing the results obtained in the manuscript with the results in these and other articles.

L81. Wherever a chemical material or equipment is mentioned, it is necessary to specify the manufacturer, city and country.

L86,175. There are no brackets.

L102-116. Section 2.3 does not say how TiO2 is formed.

L143. Figure 1 does not show how the gas was supplied for the study.

L158. In Fig.2 there is no response from TiO2. Why?

L179-188. This is a repeat of the previous text.

L232. In Fig.5c, the drift of the initial resistance (Ra) of the F/M-3 sample is observed. This needs to be explained.

As for Fig.5d,e.  A person's exhalation contains moisture in large quantities and other gases, including acetone. However, the authors only investigated the simultaneous effects of moisture and acetone. It is also desirable to conduct studies of moisture and other gases, at least ethanol, for example.

Regarding Fig.6,b. Since the influence of moisture is strong and reduces the response (Fig.5e), it is also desirable to study long-term stability with simultaneous exposure to acetone and moisture.

L260-261. Equations (1) and (2) are not equations, since they do not balance the number of electrons and atoms of the elements. This can be called the direction of the reaction. In (1) and (2), the O2-ion is present twice.

In the Discussion section, it is necessary to provide a table comparing the results obtained in the manuscript with the results in these and other articles.

L95. Fig.7 needs to be made clearer, the text should be readable.

Round 2

Reviewer 2 Report

Comments and Suggestions for Authors

In general, the authors of the manuscript responded to my comments. There were minor typos in the article, such as on L91. New articles with typos have been added to References. There is an incomprehensible sign in the links to the cited articles.
In addition, I would like to note that the moisture content in the air must be measured, since in some cases the sensor response depends on the moisture content in the air. I would like the authors to say something about this.
